# Postherpetic Pseudolymphomatous Angiosarcoma Concealed Within Milia en Plaque: Expanding the Spectrum of Wolf Isotopic Response with a Literature Review

**DOI:** 10.3390/dermatopathology12020009

**Published:** 2025-03-22

**Authors:** Marina Corral-Forteza, Noelia Pérez-Muñoz, Maria-Teresa Fernández-Figueras

**Affiliations:** 1Grupo Quironsalud, Hospital Universitari Sagrat Cor, 08029 Barcelona, Spain; marinacf1990@gmail.com; 2Grupo Quironsalud, Hospital Universitari General de Catalunya, 08195 Barcelona, Spain; noelia.perez@quironsalud.es; 3Department of Anatomic Pathology, Faculty of Medicine and Health Sciences, Campus Sant Cugat, Universitat Internacional de Catalunya, 08195 Barcelona, Spain

**Keywords:** wolf isotopic response, immunocompromised district, angiosarcoma, pseudolymphoma, comedones, milia, herpes zoster, postherpetic

## Abstract

The Wolf isotopic response (WIR) refers to the development of cutaneous lesions in areas of previously healed but unrelated skin disease. While most are observed in healed herpes zoster, WIR has been reported in various other contexts. Affected areas are believed to exhibit immune dysregulation, lymphatic dysfunction, and altered neuromediator activity, increasing susceptibility to inflammatory, neoplastic, and infectious conditions. This phenomenon aligns with the broader concept of the “immunocompromised district”, which also encompasses the Koebner phenomenon and its reverse. Herein, we present the case of a 96-year-old woman who developed multiple cysts and comedones at the site of a resolved herpes zoster. Due to persistent and refractory inflammation, curettage was performed, and histopathological examination revealed angiosarcoma with a pseudolymphomatous reaction interspersed among the cysts. The coexistence of multiple types of WIR is rare but not unprecedented, highlighting the importance of recognizing the diverse pathologic conditions that can arise in such settings. In this review, we explore the historical evolution of terminology used to describe lesions in vulnerable skin areas and related phenomena. We also provide an updated overview of current pathogenic theories and present a comprehensive compilation of postherpetic reactions reported to date.

## 1. Introduction

Wolf isotopic response (WIR) is defined as the appearance of new skin lesions at the site of a previously healed, unrelated skin disease [1]. This intriguing phenomenon was first described in 1955 by Wyburn-Mason [2], but it remained relatively unrecognized until 1995 when Wolf formally established the concept and introduced the term “isotopic response [3]”. This terminology was intended to distinguish it from the Koebner isomorphic phenomenon, in which a disease develops in a previously healthy area that has undergone inflammation due to trauma, infection, or other conditions. Until that moment, cases of WIR had traditionally been considered variants of the Koebner phenomenon [4]. The terms “isotopic” and “isomorphic” highlight the fundamental differences between these two reactions “isotopic” refers to the development of two distinct diseases at the same site, and “isomorphic” (from Greek, “equal shape”) emphasizes that the new condition reproduces the outline of the preceding lesion [5].

The classification of WIR remains a topic of ongoing debate in dermatology. Over time, a diverse and sometimes inconsistent terminology has emerged in the literature, with terms such as locus minoris resistentiae, isomorphic, isopathic response, and pseudoisomorphic response being used to describe similar phenomena [6]. Additionally, the distinction between WIR and the Koebner phenomenon has been widely questioned, with some experts strongly disputing their classification as separate entities [4].

In 2014, Ruocco et al. introduced the concept of the immunocompromised cutaneous district (ICD) to unify these terminologies and provide a broader pathophysiological framework encompassing WIR and related phenomena [6]. The ICD model emphasizes localized immune dysregulation as a key factor in creating vulnerable skin regions predisposed to secondary dermatological conditions. However, growing evidence suggests that WIR development is influenced by additional mechanisms, including structural alterations, neural dysfunction, and vascular damage [6].

A wide range of conditions has been proposed to predispose the WIR, including congenital malformations such as primary lymphedema or skin mosaicisms as well as acquired conditions such as secondary lymphedema, neural diseases, trauma, burns, tattoos, and intradermal vaccinations [6]. However, the most common underlying causes of this localized vulnerability are herpetic infections, with herpes zoster being far more frequent than herpes simplex. The link between herpetic infections and WIR is particularly significant, as these infections appear to induce long-term alterations in the cutaneous immune microenvironment. Even after the initial infection resolves, the affected skin remains in a localized immunocompromised state, making it more susceptible to the development of new, unrelated dermatological conditions.

The spectrum of WIR manifestations following herpetic infections is remarkably diverse, encompassing a variety of skin disorders such as granulomatous reactions, lichenoid eruptions, psoriasis, and even malignant transformations. Additionally, these compromised areas may exhibit an increased susceptibility to metastatic deposits or local involvement by systemic lymphoproliferative disorders. Furthermore, the heightened incidence of lesions in these compromised regions can lead to the coexistence of multiple pathologic conditions at the same site, posing significant diagnostic and therapeutic challenges for dermatopathologists and clinicians [7].

## 2. Case Report

The 96-year-old woman had a history of herpes zoster involving the right side of her face, corresponding to the second branch of the trigeminal nerve, which resolved spontaneously. Two years later, she developed numerous milia and comedones in the same area, accompanied by edema, erythema, serous transudation, and severe pain. Magnetic resonance imaging revealed microcystic subcutaneous lesions in the right infraorbital region. Based on these findings, a clinical diagnosis of inflamed milia en plaque was made. Despite treatment with topical agents and oral antibiotics, only partial improvement was achieved, necessitating surgical curettage.

Histopathological examination of the curetted material, composed of numerous skin fragments, revealed features of inflamed milia en plaque (Figure 1). In many fragments, an abundant lymphocytic infiltrate obscured a population of highly atypical cells arranged in cords and sheets that co-expressed CD31 and D2-40 (Figure 2), an immunoprofile characteristic of angiosarcoma.

Additionally, occasional cavities lined by atypical endothelial cells were observed, containing free-floating tumor aggregates with central vessels and abundant stromal lymphocytes (Figure 3). This finding corresponds to the “fish in the creek” pattern, which is also characteristic of angiosarcoma [8]. These features led to a final diagnosis of pseudolymphomatous angiosarcoma associated with multiple comedones.

Given the patient’s advanced age and her preference, conservative management with palliative care was chosen.

## 3. Discussion

The clinicopathological case we present is paradigmatic of WIR, with the coexistence of three distinct pathologic conditions at the site of a healed herpes zoster infection: angiosarcoma, pseudolymphomatous infiltrates, and comedones [9,10,11]. The occurrence of multiple synchronous WIR pathology conditions is rare but not unique, with reports documenting two [12] or even three [7] pathologic conditions at the same site. In this patient, the pseudolymphomatous infiltrates were likely related to WIR, given the well-documented inflammatory processes associated with this phenomenon. However, an alternative explanation could attribute the infiltrates to a rare pseudolymphomatous variant of angiosarcoma [13], with the lymphocytic component being primarily tumor-related rather than part of a WIR. While definitive proof is lacking, the rarity of this angiosarcoma variant and the presence of multiple simultaneous entities in the same site strongly suggest that WIR played a role in driving the inflammatory response. Conversely, comedones have not been previously described in association with angiosarcoma, which has been linked only to verrucous epidermal hyperplasia [14].

The exact etiopathogenesis of WIR remains unclear. Various factors have been proposed, and it is likely that multiple mechanisms are involved in different cases, which are listed as follows:

Viral persistence: This may explain some postherpetic reactions since viral DNA has been isolated in certain cases. However, after herpetic infections resolve, viral DNA is no longer detected. This mechanism likely applies only to reactions occurring within a short interval (less than four weeks) after the herpes infection [6,15].Lymphovascular microvasculature alteration: Damage to the microcirculation can render an area unable to respond adequately to subsequent insults, leading to localized inflammation at the same site [6].Local immune dysregulation: The initial skin disease may impair normal immunological functions in the affected area. Altered regional immunity, including the contribution of resident memory T cells, could predispose the site to new cutaneous diseases following a trigger [6,16].Neural injury: Damaged dermal nerve fibers can contribute to disease pathogenesis, either directly through neuropeptide release or indirectly via immune system activation [6]. Specifically, neuropeptides such as nerve growth factor (NGF), which regulates skin epithelization, angiogenesis, and the formation of an extracellular matrix [17], as well as substance P from damaged nerve endings, which might play a crucial role in inducing the development of epidermal changes [18].

In addition, a plausible explanation for the milia could be related to sebaceous gland lipogenesis stimulated by substance P, followed by the proliferation of Propionibacterium acnes [11].

Postherpetic reactions represent the most common scenario for WIR, and a wide spectrum of pathologic conditions have been described in this context. Ruocco and colleagues [6] collected a large series of cases, which has since been expanded with additional reports. We provide an updated list of the inflammatory, infectious, and neoplastic postherpetic reactions reported to date, arranged in alphabetical order to facilitate the search for their histopathological features and ensure none are overlooked (Table 1).

Most WIR cases involve inflammatory dermatoses, granulomatous reactions being the most frequent. The spectrum also includes infections, reactive processes such as milia or pseudolymphomas, and tumors. Among the latter, vascular lesions, including angiosarcoma, have occasionally been reported in areas previously affected by herpes zoster [9,10,19,20].

**Table 1 dermatopathology-12-00009-t001:** Postherpetic Wolf isotopic responses. Inflammatory, infectious, and neoplastic postherpetic reactions reported up to date listed in alphabetical order.

Inflammatory Diseases	Infections
Acne	Molluscum contagiosum
Acneiform lesions	Warts and papillomata
Actinic granuloma (O’Brien) [21]	Candidiasis
Bullous pemphigoid [18]	Dermatophytosis
Chronic cutaneous graft-versus-host disease	
Chronic small vessel vasculitis (Extrafacial eosinophilic granuloma or Lever granuloma) [22]	
Chronic urticaria [23]	
Comedones	
Comedonic-microcystic reactions	
Contact dermatitis	
“Dysimmune” reactions	
Eosinophilic dermatosis	
Erythema annulare centrifugum	
Fibroelastolytic papulosis	
Folliculitis (granulomatous or eosinophilic [24])	**Tumors and pseudotumors**
Follicular mucinosis [25]	Angiosarcoma
Furunculosis	Basal cell carcinoma
Granuloma annulare [26]	Basosquamous carcinoma
Granulomatous folliculitis	Benign lymphangioendothelioma [19]
Granulomatous reactions (necrotizing or non-necrotizing)	Bowen disease
Granulomatous vasculitis	Breast carcinoma
Grover disease (personal case)	Kaposi sarcoma
Infections	Leukemia
Keloid	Lymphangiogenesis (pseudotumoral) [19]
Lichen planus	Lymphomas
Lichen sclerosus et atrophicus-morphea	Mastocytosis [27]
Lichen simplex	Metastases (from breast carcinoma and others)
Lichenoid dermatitis	Pseudolymphoma
Linear IgA dermatosis	Rosai-Dorfman disease
Lupus erythematosus [28]	Squamous cell carcinoma
Milia	Syringoid eccrine carcinoma [18]
Mucinosis	Tufted Angioma [29]
Nodular solar degeneration	
Palpable purpura	
Pityriasis rosea (atypical) [30]	
Prurigo-like eruption	
Psoriasis	
Reactive perforating collagenosis	
Rosacea	
Sarcoidosis	
Unilateral nevoid telangiectasia [31]	
Vitiligo [32]	
Xanthoma [31]	

The history of WIR exemplifies the ongoing evolution of knowledge in dermatological sciences. What began as a keen clinical observation, initially imprecise but insightful, gradually evolved into a refined model of WIR, which has been expanded and redefined over time. As our understanding of the intricate relationship between skin integrity, immune function, and disease manifestations has evolved, certain subclassifications of cutaneous reactions now appear redundant. A logical approach involves categorizing these reactions based on their proposed pathophysiological mechanisms, leading to broader, more integrative concepts, such as the ICD, which provide a unifying framework for these phenomena.

This process began in 1955 when Wyburn-Mason first published a series of cases highlighting an apparent increased risk of malignant changes in skin previously affected by herpes zoster [2]. He described 10 cases on the face and 15 on the thorax. Notably, while the facial cases involved cutaneous malignancies (five basal cell carcinomas, four squamous cell carcinomas, and two cases with mixed features), only one of the thoracic cases was a cutaneous tumor (squamous cell carcinoma). The remaining 14 cases were mammary gland carcinomas that, despite involving the same breast whose overlying skin had been previously affected by herpes zoster, would not be considered WIR by the present standards, but rather a coincidental occurrence.

The term isotopic response was later coined by Wolf in 1985 to describe this phenomenon, emphasizing the development of a new, unrelated skin disorder at the exact site of a previous, healed skin condition. However, the word isotopic is also widely used in medical fields to refer to radioactive isotopes, a coincidence that may have led to confusion and difficulties in retrieving specific scientific literature on this topic. To avoid ambiguity, the eponym Wolf was added, making Wolf isotopic response (WIR) the preferred and most used term.

Today, WIR is recognized as a distinct dermatological entity, with a wide range of predisposing conditions. These include congenital malformations (such as primary lymphedema, and skin mosaicisms) and acquired conditions (such as secondary lymphedema, neural diseases, trauma, burns, tattoos, or intradermal vaccinations). Its clinical significance underscores the importance of considering the dermatological history of every patient when evaluating new skin lesions.

Distinguishing WIR and Koebner’s isomorphic response or its reverse, the so-called Renboek phenomenon, remains relevant for the sake of diagnostic precision. However, overlapping pathophysiological mechanisms likely exist [4], including increased local expression of nerve growth factor (NGF) and vascular endothelial growth factor (VEGF), which may contribute to disease susceptibility in affected areas [33]. Clinicians and dermatopathologists sometimes struggle to classify their patients’ conditions. Table 2 provides definitions and examples to aid in the traditional classification of these entities and, most importantly, to distinguish them from pathergy.

In 2014, Ruocco et al. introduced the ICD concept, providing a unifying pathophysiological framework that encompasses WIR and related conditions. However, this terminology has not yet been widely introduced into clinical practice.

## 4. Conclusions

Shifting from traditional classifications exposed in Table 2 to newer frameworks where most of these reactions are grouped under the umbrella term of the Koebner phenomenon or the newer terminology of ICD, requires time. It will be necessary to overcome inertia and dismantle long-standing paradigms. In the meantime, the specific terminology used in clinical practice is of secondary importance, as long as scientific progress continues to refine our understanding of their likely shared underlying mechanisms and ensures that therapeutic strategies remain effective and evidence-based.

The clinicopathological case included in this review exemplifies the importance of recognizing the diverse spectrum of postherpetic WIR and closely monitoring patients who develop new lesions at previously healed sites of another skin disease. This is particularly critical given the potential for malignant tumors to develop. Early histopathological studies with adequately sized specimens are crucial to enable a prompt diagnosis. In the patient we are reporting, a superficial shave biopsy would have likely confirmed the presence of milia, comedones, and lymphocytic infiltrates, but could have missed the angiosarcoma. Dermatopathologists should be familiar with the patterns of WIR and remember that some patients may develop multiple synchronous lesions requiring meticulous evaluation.

## 5. Future Directions

Recent advances in understanding the complex interplay between nerve stimuli—mediated by neuropeptides and other soluble neurotransmitters—and the dermoepidermal tissues are shedding light on wound repair processes and the pathogenesis of inflammatory cutaneous diseases. These findings are expected to provide valuable insights into the physiopathology of WIR and, more broadly, the concept of the “immunocompromised district”. Grouping phenomena such as WIR, the Koebner phenomenon, and similar reactions under this unified concept offer a cohesive framework that may facilitate further research and enhance our understanding of these processes.

## Figures and Tables

**Figure 1 dermatopathology-12-00009-f001:**
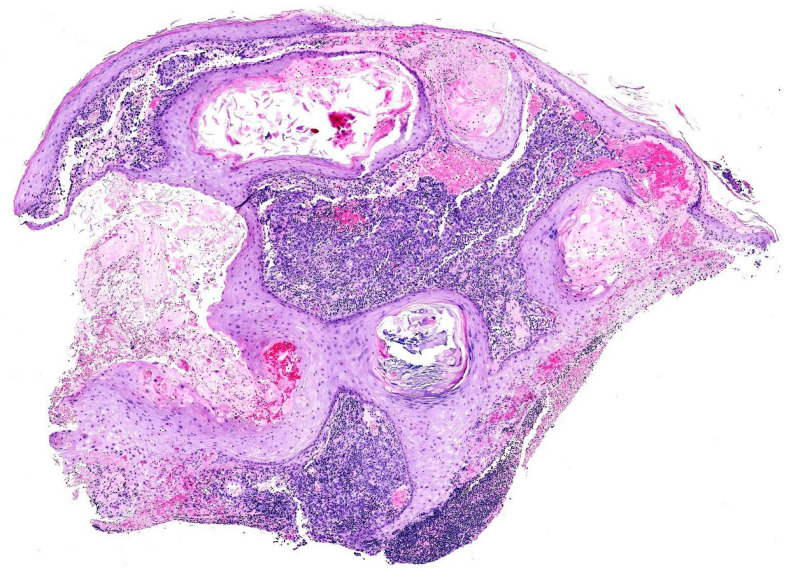
Comedones and milia cysts consistent with milia en plaque, accompanied by a dense interstitial lymphoid infiltrate. At low magnification, the interconnected vascular channels and solid areas of angiosarcoma, located in the interstitial spaces, can be easily overlooked.

**Figure 2 dermatopathology-12-00009-f002:**
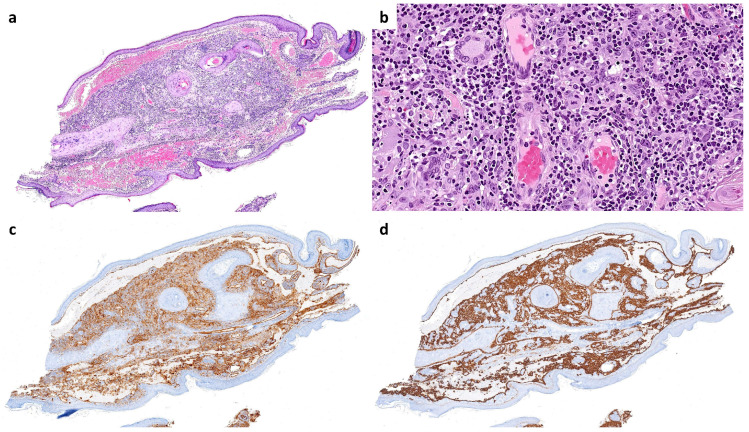
In many fragments, a population of atypical cells with occasional mitoses was observed forming cords and loose masses, obscured by the lymphoid infiltrate (**a**,**b**). These cells showed immunohistochemical positivity for CD31 (**c**) and D2-40 (**d**).

**Figure 3 dermatopathology-12-00009-f003:**
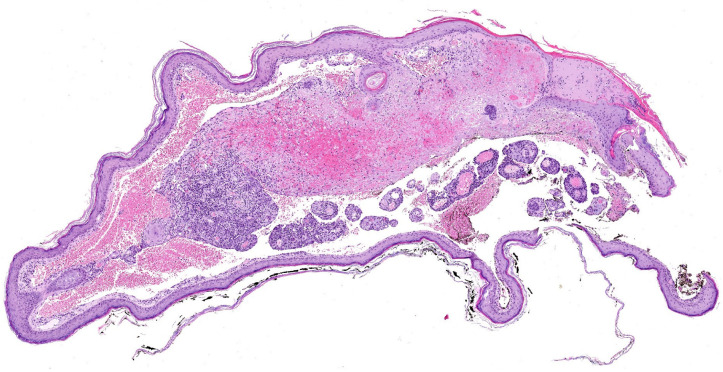
A large cavity in one fragment is lined with atypical endothelial cells and contains free-floating tumor aggregates. Each aggregate presents a central vessel, is surrounded by stroma rich in lymphocytes, and is covered by endothelial cells at the periphery. This finding, often compared to the “fish-in-the-creek” pattern, is characteristic of angiosarcoma.

**Table 2 dermatopathology-12-00009-t002:** Terminology for lesions developed in vulnerable areas of the skin and similar responses. Definitions and examples of different cutaneous reactions in vulnerable areas of the skin and similar responses, with information about the first description and some interesting facts marked with an *.

Term	First Described	Definition and * Interesting Facts	Example
**Koebner isomorphic phenomenon** [4,5]	H. Koebner, 1872	Development of lesions from an existing cutaneous disease in previously healthy skin that has undergone non-specific injury, which can be mild. This disease may appear in locations where it typically does not occur.	Psoriasis lesions developing on surgical scars in a patient with underlying psoriasis.
**Wolf isotopic response [2,3]**	R. Wyburn-Mason, 1955; refined by R. Wolf, 1995	Appearance of new skin lesions at the site of a previously healed, unrelated skin disease.	Comedones appearing exactly at the site of a previously healed herpes zoster.
***Locus minoris resistentiae* [34,35]**	R.L. Zuehlke, 1982	Areas of the body that are more vulnerable than others to suffer some pathologic conditions. * Its origins trace back to ancient myths, such as Achilles’ heel or Siegfried’s shoulder.	Eczema appearing in skin previously damaged by surgery or burns.
**Isopathic phenomenon [36]**	F. Sagher, 1954	The injection of specific or non-specific proteins triggers the development of a disease already present in the patient.	In leprous patients, the cutaneous injection of several substances provokes changes typical of lepromatous leprosy. Conversely, in healthy individuals, the same injections cause only nonspecific inflammatory reaction. Similar reactions can be triggered by insect bites.
**Pseudoisomorphic response (pseudo-Koebner) [37,38,39]**	Unknown. First examples in the literature from F. Lewandowsky, W. Lutz, 1922	Spread of a cutaneous infection along a line of previously damaged skin.	Lineal lesions in epidermodysplasia verruciformis. Commonly seen in warts and molluscum contagiosum.
**Renboek/Renbök phenomenon (inverse or reverse Koebner phenomenon) [40,41]**	R. Happle, 1991	Disappearance of an existing skin condition following the onset of a new dermatosis at the same site. * The term “Renboek” or “Renbök” is not an eponym, but “Koebner” spelled backwards.	A plaque of scalp psoriasis disappears as a patch of alopecia areata develops in the same location.
**Pathergy [29]**	Blobner, 1937	A minor trauma on healthy skin induces non-specific cutaneous lesions, often rich in neutrophils.	Frequently seen in pyoderma gangrenosum, Sweet’s syndrome or Beçet’s disease.

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
