# Peer review of "Postherpetic Pseudolymphomatous Angiosarcoma Concealed Within Milia en Plaque: Expanding the Spectrum of Wolf Isotopic Response with a Literature Review"

_dermatopathology, 2025, doi:10.3390/dermatopathology12020009_

Round 1

Reviewer 1 Report

Comments and Suggestions for Authors

The authors describe a very interesting case report of a patient developing a pseudolymphomatous angiosarcoma within milia en plaque at the site of previous herpes zoster (expanding the spectrum of entities that may be involved in Wolf isotopic response). The introduction and discussion are extremely helpful and provide a nice literature review about these concepts.

The case presentation is rather unique and amazing, and it is full of potential pitfalls of particular interest for pathologists in general and dermatopathologists in particular.

The pictures are of good quality and the figure legends are relevant and accurate.

  • In the case presentation, the authors state that the patient had been previously treated with topical "agents"; if possible, please clarify which types of topical treatments were previously tried.
  • If available, a clinical picture would also enrich the case report.

Author Response

The authors describe a very interesting case report of a patient developing a pseudolymphomatous angiosarcoma within milia en plaque at the site of previous herpes zoster (expanding the spectrum of entities that may be involved in Wolf isotopic response). The introduction and discussion are extremely helpful and provide a nice literature review about these concepts.

The case presentation is rather unique and amazing, and it is full of potential pitfalls of particular interest for pathologists in general and dermatopathologists in particular.

The pictures are of good quality and the figure legends are relevant and accurate.

  • In the case presentation, the authors state that the patient had been previously treated with topical "agents"; if possible, please clarify which types of topical treatments were previously tried.
  • If available, a clinical picture would also enrich the case report.

Response:

We sincerely appreciate the positive evaluation of our paper.

Regarding the reviewer’s requests:

  • Unfortunately, no photographs were taken of the patient, since the condition was initially considered a non-specific process.
  • As for the topical treatment administered a few weeks before the biopsy, it was mentioned but not specified. While it was likely a topical corticosteroid, we chose not to assume this without definitive confirmation.  

Reviewer 2 Report

Comments and Suggestions for Authors

This is a highly valuable contribution that provides detailed insights into Wolf’s phenomenon.

The paper describes a case of a complex isotopic response and gives a comprehensive overview on related phenomenons.

The results are thoroughly analyzed, discussed, and, excellently illustrated.

Author Response

This is a highly valuable contribution that provides detailed insights into Wolf’s phenomenon.

The paper describes a case of a complex isotopic response and gives a comprehensive overview on related phenomenons.

The results are thoroughly analyzed, discussed, and, excellently illustrated.

Response:

We truly appreciate the positive evaluation of our paper.  

Reviewer 3 Report

Comments and Suggestions for Authors

First of all, in my opinion, is important to unify different terminologies and to make a great effort to provide one affordable pathophysiological concept. There are various factors proposed for the etiopathogenesis of WIR and this effort could explain  the variety of simultaneous entities in the same lesion.. Comedones has been described after  herpetic infections and perhaps could be related to a proliferation of Propionibacterium and not to a neural injury or lymphovascular alteration. Nevertheless must to be considered a possible complication ,not rare, of these viral infections.The presence of BCC and SCC and two lesions with mixed features in the face should underline simply the coexistence of sun action in  immunocompromised site. To underline again is the history of patients disease and depth of skin biopsy to give to a dermatopathologist  the right chance of diagnosis and the impulse to better understanding the pathogenesis.

Author Response

Reviewer comments

First of all, in my opinion, is important to unify different terminologies and to make a great effort to provide one affordable pathophysiological concept. There are various factors proposed for the etiopathogenesis of WIR and this effort could explain  the variety of simultaneous entities in the same lesion.. Comedones has been described after  herpetic infections and perhaps could be related to a proliferation of Propionibacterium and not to a neural injury or lymphovascular alteration. Nevertheless must to be considered a possible complication ,not rare, of these viral infections. The presence of BCC and SCC and two lesions with mixed features in the face should underline simply the coexistence of sun action in  immunocompromised site. To underline again is the history of patients disease and depth of skin biopsy to give to a dermatopathologist  the right chance of diagnosis and the impulse to better understanding the pathogenesis.

Response:  

We sincerely appreciate the reviewer’s thoughtful insights on our paper.

  • We fully agree on the importance of unifying terminologies. In our opinion, the concept of the immunocompromised district provides an ideal framework for a pathophysiological explanation, as we mention in lines 227–229: “Grouping phenomena such as WIR, the Koebner phenomenon, and similar reactions under this unified concept offers a cohesive framework that may facilitate further research and enhance our understanding of these processes.” As the reviewer notes, the presence of BCC, SCC, and lesions with mixed features on the face likely reflects the combined effects of sun exposure in an immunocompromised site. Nevertheless, we acknowledge that these entities can also be encompassed under the broader concept of Wolf’s isotopic response, a term more traditionally used in medical literature and widely recognized in clinical practice. Additionally, leading experts (reference 4) have advocated for this terminology, which we have adopted in our paper.
  • We also agree that multiple factors may contribute to the pathogenesis of WIR. As stated in lines 122–123: “The exact etiopathogenesis of WIR remains unclear. Various factors have been proposed, and it is likely that multiple mechanisms are involved in different cases.” Furthermore, we have included a discussion on the possible role of Propionibacterium acnes in our manuscript (lines 141–143): “In addition, a plausible explanation for the milia could be related to sebaceous gland lipogenesis stimulated by substance P, followed by the proliferation of Propionibacterium acnes.”